# Walking the Tightrope: An Investigation of the Convolutional Autoencoder Bottleneck

## Abstract

In this paper, we present an in-depth investigation of the convolutional autoencoder (CAE) bottleneck. Autoencoders (AE), and especially their convolutional variants, play a vital role in the current deep learning toolbox. Researchers and practitioners employ CAEs for a variety of tasks, ranging from outlier detection and compression to transfer and representation learning. Despite their widespread adoption, we have limited insight into how the bottleneck shape impacts the emergent properties of the CAE. We demonstrate that increased height and width of the bottleneck drastically improves generalization, which in turn leads to better performance of the latent codes in downstream transfer learning tasks. The number of channels in the bottleneck, on the other hand, is secondary in importance. Furthermore, we show empirically that, contrary to popular belief, CAEs do not learn to copy their input, even when the bottleneck has the same number of neurons as there are pixels in the input. Copying does not occur, despite training the CAE for 1,000 epochs on a tiny ($\approx 600$ images) dataset. We believe that the findings in this paper are directly applicable and will lead to improvements in models that rely on CAEs.

## 1 Introduction

Autoencoders (AE) are an integral part of the neural network toolkit. They are a class of neural networks that consist of an encoder and decoder part and are trained by reconstructing datapoints after encoding them. Due to their conceptual simplicity, autoencoders often appear in teaching materials as introductory models to the field of deep unsupervised learning. Nevertheless, autoencoders have enabled major contributions in the application and research of the field. The main areas of application include outlier detection (Xia et al., 2015; Chen et al., 2017; Zhou & Paffenroth, 2017; Baur et al., 2019), data compression (Yildirim et al., 2018; Cheng et al., 2018; Dumas et al., 2018), and image enhancement (Mao et al., 2016; Lore et al., 2017). In the early days of deep learning, autoencoders were a crucial tool for the training of deep models. Training large (by the standards of the time) models was challenging, due to the lack of big datasets and computational resources. One way around this problem was to pre-train some or all layers of the network greedily by treating them as autoencoders with one hidden layer (Bengio et al., 2007). Subsequently, Erhan et al. (2009) demonstrated that autoencoder pre-training also benefits generalization. Currently, researchers in the field of representation learning frequently rely on autoencoders for learning nuanced and high-level representations of data (Kingma & Welling, 2013; Tretschk et al., 2019; Shu et al., 2018; Makhzani et al., 2015; Berthelot et al., 2018).

However, despite its widespread use, we propose that the (deep) autoencoder model is not well understood. Many papers have aimed to deepen our understanding of the autoencoder through theoretical analysis (Nguyen et al., 2018; Arora et al., 2013; Baldi, 2012; Alain & Bengio, 2012). While such analyses provide valuable theoretical insight, there is a significant discrepancy between the theoretical frameworks and actual behavior of autoencoders in practice, mainly due to the assumptions made (e.g., weight tying, infinite depth) or the simplicity of the models under study. Others have approached this issue from a more experimental angle (Arpit et al., 2015; Bengio et al., 2013; Le, 2013; Vincent et al., 2008; Berthelot et al., 2019). Such investigations are part of an ongoing effort to understand the behavior of autoencoders in a variety of settings.

The focus of most such investigations so far has been the traditional autoencoder setting with fully connected layers. When working with image data, however, the default choice is to use convolutions, as they provide a prior that is well suited to this type of data (Ulyanov et al., 2018). For this reason, Masci et al. (2011) introduced the convolutional autoencoder (CAE) by replacing the fully connected layers in the classical AE with convolutions. In an autoencoder, the layer with the least amount of neurons is referred to as a bottleneck. In the regular AE, this bottleneck is simply a vector ( rank-1 tensor). In CAEs, however, the bottleneck assumes the shape of a multichannel image (rank-3 tensor, height × width × channels) instead of a vector. This bottleneck shape prompts the question: What is the relative importance of the number of channels versus the height and width (hereafter referred to as size) in determining the tightness of the CAE bottleneck? Intuitively, we might expect that only the total number of neurons should matter since convolutions with one-hot filters can distribute values across channels. Generally, the study of CAE properties appears to be underrepresented in literature, despite their widespread adoption.

In this paper, we share new insights into the properties of convolutional autoencoders, which we gained through extensive experimentation. We address the following questions:

- How does the number of channels and the feature map size in the bottleneck layer impact
  - reconstruction quality?
  - generalization ability?
  - the structure of the latent code?
  - knowledge transfer to downstream tasks?
- How and when do CAEs overfit?
- How does the complexity of the data distribution affect all of the above?
- Are CAEs capable of learning a "copy function" if the CAE is complete (i. e., when the number of pixels in input equals the number of neurons in bottleneck)? This "copying CAE" hypothesis is a commonly held belief that was carried over from regular AEs (see Sections 4 and 5 in Masci et al. (2011).

We begin the following section by formally introducing convolutional autoencoders and explaining the convolutional autoencoder model we used in our experiments. Additionally, we introduce our three datasets and the motivation for choosing them. In Section 3, we outline the experiments and their respective aims. Afterward, we present and discuss our findings in Section 4. All of our code, as well as the trained models and datasets, will be published at https://github.com/YmouslyAnon/WalkingTheTightrope. This repository will also include an interactive Jupyter Notebook for investigating the trained models. We invite interested readers to take a look and experiment with our models.

## 2 MATERIALS AND METHODS

### 2.1 AUTOENCODERS AND CONVOLUTIONAL AUTOENCODERS

The regular autoencoder, as introduced by Rumelhart et al. (1985), is a neural network that learns a mapping from data points in the input space $\boldsymbol{x} \in \mathbb{R}^d$ to a code vector in latent space $\boldsymbol{h} \in \mathbb{R}^m$ and back. Typically, unless we introduce some other constraint, $m$ is set to be smaller than $d$ to force the autoencoder to learn higher-level abstractions by having to compress the data. In this context, the encoder is the mapping $f(x) : \mathbb{R}^d \rightarrow \mathbb{R}^m$ and the decoder is the mapping $g(h) : \mathbb{R}^m \rightarrow \mathbb{R}^d$. The layers in both the encoder and decoder are fully connected:

$$\boldsymbol{l}^{i+1} = \sigma(\boldsymbol{W}^i \boldsymbol{l}^i + \boldsymbol{b}^i). \tag{1}$$

Here, $\boldsymbol{l}^i$ is the activation vector in the i-th layer, $\boldsymbol{W}^i$ and $\boldsymbol{b}^i$ are the trainable weights and $\sigma$ is a element-wise non-linear activation function. If necessary, we can tie weights in the encoder to the ones in the decoder such that $\boldsymbol{W}^i = (\boldsymbol{W}^{n-i})^T$, where $n$ is the total number of layers. Literature refers to autoencoders with this type of encoder-decoder relation as weight-tied.

The convolutional autoencoder keeps the overall structure of the traditional autoencoder but replaces the fully connected layers with convolutions:

$$\mathbf{L}^{i+1} = \sigma(\mathbf{W}^i * \mathbf{L}^i + \boldsymbol{b}^i), \tag{2}$$

where $*$ denotes the convolution operation and the bias $\boldsymbol{b}^i$ is broadcast to match the shape of $\mathsf{L}^i$ such that the j-th entry in $\boldsymbol{b}^i$ is added to the j-th channel in $\mathsf{L}^i$. Whereas before the hidden code was an m-dimensional vector, it is now a tensor with a rank equal to the rank of the input tensor. In the case of images, that rank is three (height, width, and the number of channels). CAEs generally include pooling layers or convolutions with strides $> 1$ or dilation $> 1$ in the encoder to reduce the size of the input. In the decoder, unpooling or transposed convolution layers (Dumoulin & Visin, 2016) inflate the latent code to the size of the input.

## 2.2 OUR MODEL

Our model consists of five strided convolution layer in the encoder and five up-sampling convolution layers (bilinear up-sampling followed by padded convolution) (Odena et al., 2016) in the decoder. We chose to use five layers so that the size of the latent code, after the strided convolutions, would be 4x4 or 3x3 depending on the dataset. To increase the level of abstraction in the latent code, we increased the depth of the network by placing two residual blocks (He et al., 2016) with two convolutions each after each every strided / up-sampling convolution layer. We applied instance normalization (Ulyanov et al., 2016) and ReLU activation (Nair & Hinton, 2010) following every convolution in the architecture.

One of our goals was to understand the effect latent code shape has on different aspects of the network. Therefore, we wanted to be able to change the shape of the bottleneck from one experiment to another, while keeping the rest of the network constant. To this end, we quadrupled the number of channels with every strided convolution $s^i$ and reduced it by a factor of four with every up-sampling convolution $u^i$. In effect, this means that the volume (i. e., height $\times$ width $\times$ channels) of the feature maps is identical to the input in all layers up to the bottleneck:

$$s^i(\mathsf{L}^i) \in \mathbb{R}^{h^i/2 \times w^i/2 \times 4n_c^i} \text{ , for } \mathsf{L}^i \in \mathbb{R}^{h^i \times w^i \times n_c^i} \tag{3}$$

$$u^i(\mathsf{L}^i) \in \mathbb{R}^{2h^i \times 2w^i \times n_c^i/4} \text{ , for } \mathsf{L}^i \in \mathbb{R}^{h^i \times w^i \times n_c^i} \tag{4}$$

In this regard, our model, differs from CAEs commonly found in literature, where it is customary to double/halve the number of channels with every down-/up-sampling layer. However, our scheme allows us to test architectures with different bottleneck shapes while ensuring that the volume of the feature maps stays the same as the input until the bottleneck. In this sense, the bottleneck is the only moving part in our experiments. The resulting models range from having $\sim 50M$ to $90M$ parameters.

## 2.3 DATASETS

To increase the robustness of our study, we conducted experiments on three different datasets. Additionally, the three datasets allowed us to address the question, how the difficulty of the data (i. e., the complexity of the data distribution) affects learning in the CAE. To study this effect, we decided to run our experiments on three datasets of varying difficulty. We determined the difficulty of each dataset based on intuitive heuristics. In the following, we present the datasets in the order of increasing difficulty and our reasoning for the difficulty grading.

### 2.3.1 POKEMON

The first dataset is a blend of the images from "Pokemon Images Dataset"[1] and the type information from "The Complete Pokemon Dataset"[2], both of which are available on Kaggle. Our combined dataset consists of 793 256$\times$256 pixel images of Pokemon and their primary and secondary types as labels. To keep the training time within acceptable bounds, we resized all images to be 128$\times$128 pixels. We chose this dataset primarily for its clear structure and simplicity. The images depict only the Pokemon without background, and each image centers on the Pokemon it is showing. Additionally, the variation in poses and color palettes is limited in the images, and each image contains large regions of uniform color. Due to the above reasons and its small size, we deemed this dataset to be the "easy" dataset in our experiments. We trained our models on the first 80% of images and reserved the rest for testing.

---

[1]https://www.kaggle.com/kvpratama/pokemon-images-dataset
[2]https://www.kaggle.com/rounakbanik/pokemon

### 2.3.2 CELEBA

A step up from the Pokemon dataset in terms of difficulty is the CelebA faces dataset (Liu et al., 2015). This dataset is a collection of celebrity faces, each with a 40-dimensional attribute vector (attributes such as smiling/not smiling, male/female) and five landmarks (left and right eye, nose and left and right corner of the mouth). To be able to observe overfitting behavior, we used only the first 10,000 images in the dataset for training and the last 2,000 images for testing. Since the images also contain backgrounds of varying complexity, we argue that this leads to more complex data distribution. Furthermore, the lighting conditions, quality, and facial orientation can vary significantly in the images. However, some clear structure is still present in this dataset, as the most substantial portion of each image shows a human face. For those reasons, we defined this dataset to have "medium" difficulty. For our purposes, we resized the images to be 96×96 pixels. The original size was 178×218 pixels.

### 2.3.3 STL-10

For our last dataset, we picked STL-10 (Coates et al., 2011). This dataset consists of 96×96 pixel natural images and is divided into three splits: 5,000 training images (10 classes), 8,000 test images (10 classes), 100,000 unlabeled images. The unlabeled images also include objects that are not covered by the ten classes in the training and test splits. Analogously to CelebA, we used the first 10,000 images from the unlabeled split for training and the last 2,000 for testing of the CAE. In the experiments regarding knowledge transfer (see Section 3.2), we used all 8,000 labeled images from the test split of the dataset. As the images in this dataset show many different scenes, from varying viewpoints and under a multitude of lighting conditions, we find samples from this dataset to be the most complex and, therefore, the most difficult of the three.

## 3 EXPERIMENTS

### 3.1 AUTOENCODER TRAINING

The first experiment we conducted, and which forms the basis for all subsequent experiments, consists of training of autoencoders with varying bottleneck sizes and observing the dynamics of their training and test losses. This experiment probes the relative importance of latent code size versus its number of channels. Additionally, it was meant to provide insight into how and when our models overfit and if the data complexity (see Section 2.3) plays a discernible role in this. We also tested the widespread hypothesis that autoencoders learn to "copy" the input if there is no bottleneck. For each dataset (as introduced in Section 2.3), we selected three latent code sizes (=height=width) $s_i$, $i \in \{1, 2, 3\}$ as

$$s_i = \frac{s_{input}}{2^{n_l - i + 1}} \qquad \text{with } i \in \{1, 2, 3\}, \ n_l = 5 \tag{5}$$

In this equation, $n_l = 5$ is the number of strided convolutions in the network, and $s_{input}$ is the height (= width) of the images in the dataset. Throughout the rest of the paper, we mean width and height when we refer to the size of the bottleneck. To obtain latent codes with size $s_2$ ($s_3$), we changed the strides in the last (two) strided convolution layer(s) from two to one. For each size we then fixed four levels of compression $c_j \in \{^1/_{64}, ^1/_{16}, ^1/_4, 1\}$ and calculated the necessary number of channels $n_{c_j}$ according to

$$n_{c_j} = \frac{c_j s_{input}^2 n_{c_{input}}}{s_i^2} \qquad \text{with } i \in \{1, 2, 3\}, \ j \in \{1, 2, 3, 4\} \tag{6}$$

Here, $n_{c_{input}}$ is the number of channels in the input image. This way, the autoencoders had the same number of parameters in all layers except the ones directly preceding and following the bottleneck. We used mean squared error (MSE) between reconstruction and input as our loss function. After initializing all models with the same seed, we trained each for 1,000 epochs and computed the test error after every epoch. We repeated this process for two different seeds and used the models from the first seed in further experiments.

## 3.2 Knowledge Transfer

Another goal of our investigation was to estimate the effect of the latent code shape on transferability. Here, our idea was to train a logistic regression on latent codes to predict the corresponding labels for each dataset. Since logistic regression can only learn linear decision boundaries, this approach allows us to catch a glimpse of the sort of knowledge present in the latent code and its linear separability. Furthermore, this serves as another test for the "copying" hypothesis. If the encoder has indeed learned to copy the input, the results of the logistic regression will be the same for the latent codes and the input images. In the first step, we exported all latent codes for the training and testing data from the Pokemon and CelebA datasets. For STL-10, we extracted the latent codes for the test split since we trained on the unlabeled split, where no labels are available. In the case of CelebA, we additionally trained linear regression models to predict the facial landmarks provided in the dataset. For every autoencoder setting, we used fivefold cross-validation to strengthen the reliability of the results. We trained the linear models for 200 epochs (50 epoch in the case of CelebA landmarks) with a weight decay of 0.01 and a learning rate of $c_j/64$ (referring to Section 2.2). Besides, we also trained models directly on the image data for every dataset to serve as a baseline for comparison.

## 3.3 Pair-wise Representation Similarity

In our final experiment, we used the recently published singular vector canonical correlation analysis (SVCCA) (Raghu et al., 2017) technique to gauge the pair-wise similarity of the learned latent codes. SVCCA takes two sets of neuron activations of the shape number of neurons × data points and estimates aligned directions in both spaces that have maximum correlation. First, SVCCA calculates the top singular vectors that explain 99% of the variance using singular value decomposition (SVD). Subsequently, SVCCA finds affine transformations for each set of singular vectors that maximize their alignment in the form of correlation. Lastly, it averages the correlation for each direction in the discovered subspace to produce a scalar similarity score. In convolutional neural networks, this computation can become prohibitively expensive, due to the large size of the feature maps. For such cases, the Raghu et al. (2017) recommend transforming the feature maps using discrete Fourier transformation (DFT). In the publication, the authors show that DFT leaves SVCCA invariant (if the dataset is translation invariant) but results in a block diagonal matrix, which enables exact SVCCA computation by computing SVCCA for each neuron at a time. Additionally, they recommend down-sampling bigger feature maps in Fourier space when comparing them to smaller ones. In this experiment, we investigated the effect of latent code shape on its structure and content.

## 4 Results and Discussion

Looking at the error curves for the CAEs (Fig. 1), we make several observations:

1. The total amount of neurons in the bottleneck does not affect training as much as expected. All CAEs converge to a similar training error. We find this unexpected, as the smallest bottlenecks have only 1.56% of total neurons compared to the largest ones. Although the final differences in training error are small, we discover that the size of the bottleneck feature maps has a more substantial effect on training error than the number of channels. The larger the bottleneck width and height, the lower the training error. An interesting outlier presents itself in the plots for the Pokemon dataset. Here, we see that late in the training of the CAE with the 8x8x48 bottleneck training error suddenly spikes. At the same time, the test error drops significantly approximately to the same level as the training error. We verified that this was not due to an unintended interruption in training, by retraining the model with the same seed and obtained an identical result. Currently, it is unclear to us how such a drastic change in model parameters came about at such a late stage in training. Usually, we expect the loss landscape to become smoother the longer we train a model (Goodfellow et al., 2014). Whether this outlier is a fluke or has implications for the loss landscape of CAEs remains to be seen as our understanding of the training dynamics of neural networks deepens.

2. We observe that bottleneck shape critically affects generalization. Increasing the number of channels in the bottleneck layer seems to improve test error only slightly and not in all cases. The relationship between bottleneck size and test error, on the other hand, is clear

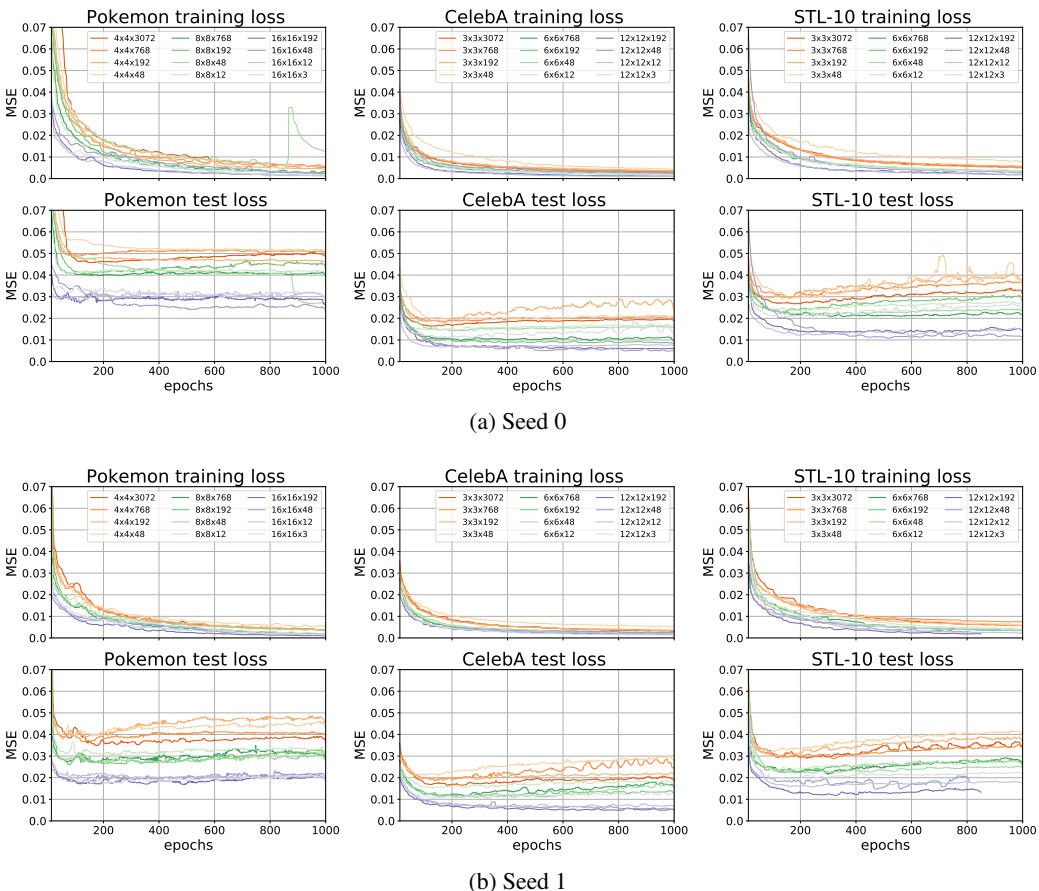

Figure 1: Loss plots for the three datasets for both seeds. Each columns corresponds to a dataset. From left to right: a) Pokemon, b) CelebA, c) STL-10. The top polot shows the training error, while the bottom one depicts test error. Every bottleneck configuration is shown as a distinct line. Configurations that have a common feature map size share the same color. Color intensity represents the amount of channels in the bottleneck (darker = more channels)

cut. Larger bottleneck size correlates with a significant decrease in test error. This finding is surprising, given the hypothesis that only the total amount of neurons matters. The CAE reconstructions further confirm this hypothesis. We visually inspected the reconstructions of our models (samples are shown in Fig. 2 and in the Appendix) and found that reconstruction quality improves drastically with the size of the bottleneck, yet no so much with the number of channels. As expected from the loss plots, the effect is more pronounced for samples from the test data.

3. Bottleneck shape also affects overfitting dynamics. We would expect the test score to increase after reaching a minimum, as the CAE overfits the data. Indeed, we observe this behavior in some cases, especially in CAEs with smaller bottleneck sizes or the minimum amount of channels. In other cases, predominantly in CAEs with a larger bottleneck size, the test error appears to plateau instead. In the plot for the CelebA dataset, the curves for 12x12x48 and 12x12x192 even appear to decrease slightly over the full training duration. This overfitting behavior implies that CAEs with a larger bottleneck size can be trained longer before overfitting occurs.

4. CAEs, where the total number of neurons in the bottleneck is the same as the number of pixels in the input, do not show signs of simply copying images. If the CAEs would indeed copy images, the test error would go to zero, yet we do not observe this case in any of the datasets. What is more, these complete CAEs follow the same pattern as the under-

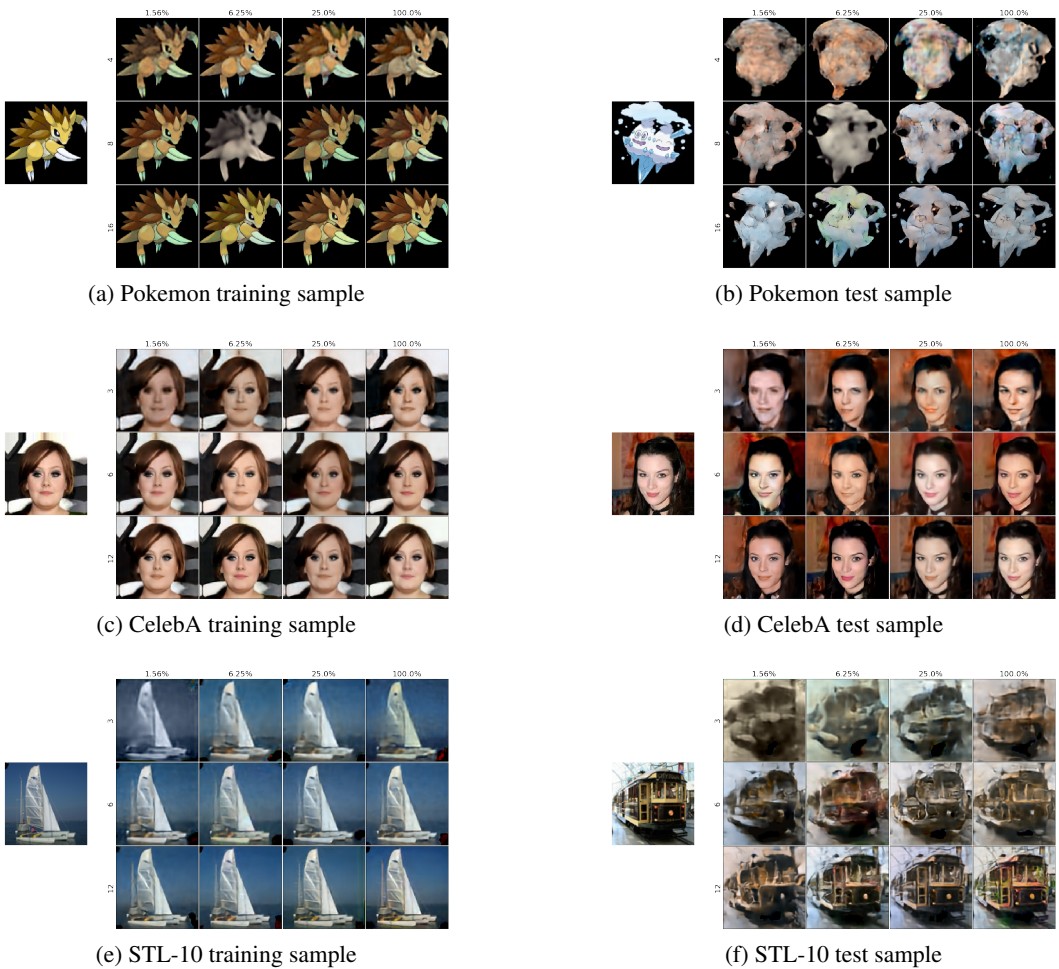

Figure 2: Reconstructions of randomly picked samples. The left column contains samples from the training data, while on the right, we show samples from the test data. In each subfigure, the rows correspond to CAEs with the same bottleneck size (height, width), increasing from top to bottom. The columns group CAEs by the number of channels in the bottleneck, expressed as percentage relative to input given bottleneck size. The image to the left of each grid is the input image.

complete ones and often converge to similar values. This finding directly contradicts the popular hypothesis about copying CAEs. In essence, it suggests that even complete CAEs learn abstractions from data, and raises the question: What prevents the CAE from simply copying its input? We believe that the answer to this question could potentially lead to new autoencoder designs that exploit this limitation to learn better representations. Hence, we argue that it is an exciting direction for future research. Additionally, the trends we derive from our results suggest that this finding likely extends to over-complete CAEs as well. However, experiments with over-complete CAEs are required to test this intuition.

Furthermore, the loss curves and reconstruction samples appear to only marginally reflect the notion of dataset difficulty, as defined in Section 2.3. One thing that stands out is the large generalization gap on the Pokemon dataset, which is most likely due to the comparatively tiny dataset size of $\approx$ 600 training images. Comparing the results for CelebA and STL-10, we find that overall generalization appears to be slightly better for CelebA, which is the less difficult dataset of the two. The test errors on STL-10 exhibit greater variance than on CelebA, although the number of samples and training epochs are equal between the two. This effect also shows itself in the reconstruction quality. On CelebA, even the CAEs with the smallest bottlenecks manage to produce decent reconstructions

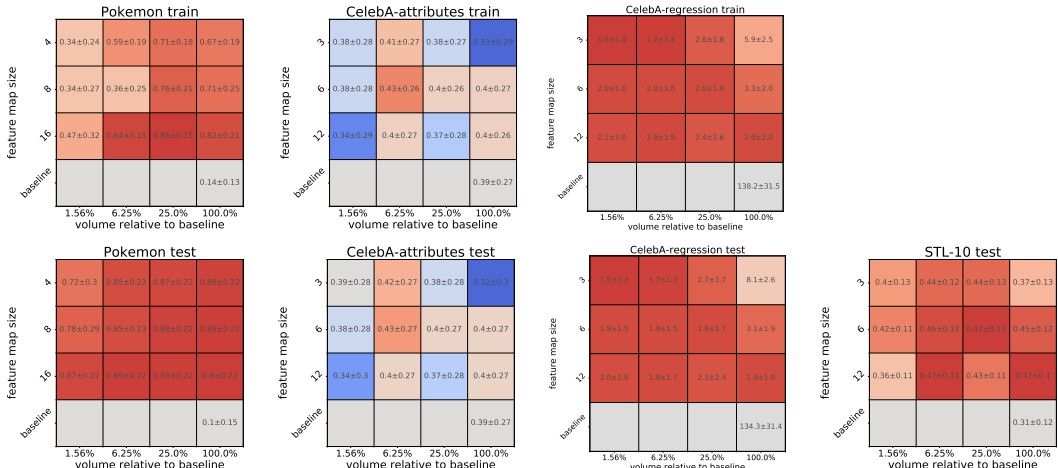

Figure 3: Results from training linear models on latent codes to predict the labels associated with each dataset. For Pokemon, CelebA attributes and STL-10 (macro) f1-score is shown. The plots for CelebA regression show MSE. The top row corresponds to models trained on latent codes from the CAE training data, while the bottom row is from CAE test data. Color is based on difference to baseline, where red signifies an improvement.

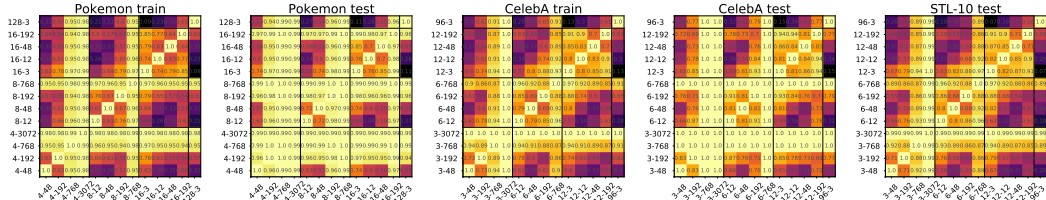

Figure 4: Results of pair-wise SVCCA. Labels on the x and y axis correspond to (height=width)-(number of channels) in the bottleneck.

on test data, whereas the test sample reconstructions on STL-10 are often unrecognizable for those models. Overall, this effect is weak and warrants a separate investigation of the relationship between data complexity and CAE characteristics, especially in the light of compelling results from curriculum learning research (Bengio et al., 2009).

If we look at the results of our knowledge transfer experiments (Fig. 3), we find further evidence that contradicts the copying autoencoder hypothesis. Although the loss curves and reconstructions already indicate that the CAE does not copy its input, the possibility remains that the encoder distributes the input pixels along the channels but the decoder is unable to reassemble the image. Here, we see that the results from the linear model trained on latent codes perform drastically better, than the ones trained on the inputs (marked "baseline" in the figure). The only deviation from this pattern seems to be the prediction of attributes on the CelebA dataset, where the performance is more or less the same for all settings. However, the prediction of landmarks on the same dataset strongly favors latent codes over raw data. As such, it seems implausible to assume that the encoder copied the input to the bottleneck. Overall, we find that knowledge transfer also seems to work better on latent codes with greater size, although the effect is not as distinct as in the loss curves.

Another point of interest to us is the discrepancy between models trained on the CAE training and test data from the Pokemon dataset. Oddly, the linear models perform better on the test data, despite the evident overfitting of the CAEs as seen in the reconstructions and loss curves. This discrepancy raises the question if overfitting happens mostly in the decoder, while the encoder retains most of its generality. We believe that this question warrants further investigation, especially in light of the recent growth in the popularity of transfer learning methods.

We notice that the latent codes from bottlenecks with the same size have higher SVCCA similarity values, as can be seen in Fig. 4 in the blocks on the diagonal. This observation further supports our hypothesis that latent code size, and not the number of channels, dictates the tightness of the CAE bottleneck. Finally, we wish to point out some observations in the SVCCA similarities as a possible inspiration for future research:

- Overall, similarity appears to be higher in latent codes from test data than in codes from training data
- Latent codes from complete CAEs show high similarity to all latent codes from all other CAEs
- SVCCA similarity with the raw inputs tends to increase with the number of channels

## 5 CONCLUSION

In this paper, we presented the findings of our in-depth investigation of the CAE bottleneck. The intuitive assumption that its total amount of neurons characterizes the CAE bottleneck could not be confirmed. We demonstrate that the height and width of the feature maps in the bottleneck are what defines its tightness, while the number of channels plays a secondary role. Larger bottleneck size (i. e., height and width) is also critical in achieving better generalization as well as a lower training error. Furthermore, we could not confirm the commonly held belief that complete CAE (i. e., CAEs with the same number of neurons in the bottleneck as pixels in the input) will learn to copy its input. On the contrary, even complete CAEs appear to follow the same dynamics of bottleneck size, as stated above. In knowledge transfer experiments, we have also shown that CAEs that overfit retain good predictive power in the latent codes, even on unseen samples. These insights are directly transferable to the two main areas of application for CAEs, outlier detection and compression/denoising: In the case of outlier detection, the model should yield a high reconstruction error on out-of-distribution samples. Using smaller bottleneck sizes to limit generalization could prove useful in this scenario. Compression and denoising tasks, on the other hand, seek to preserve image details while reducing file size and discarding unnecessary information, respectively. In this case, a bigger bottleneck size is preferable, as it increases reconstruction quality at the same level of compression.

Our investigation yielded additional results that spark new research questions. Data complexity, as estimated by human intuition, did not lead to significant differences in the training dynamics of our models. On the flipside, curriculum learning, which rests on a similar notion of difficulty, has been shown to lead to improvements in the training of classifiers and segmentation networks. The link between those two empirical results is still unclear. Another interesting question that arose from our experiments is how overfitting manifests itself in CAEs. Does it occurs mainly in the encoder or decoder or equally in both?

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

# A APPENDIX

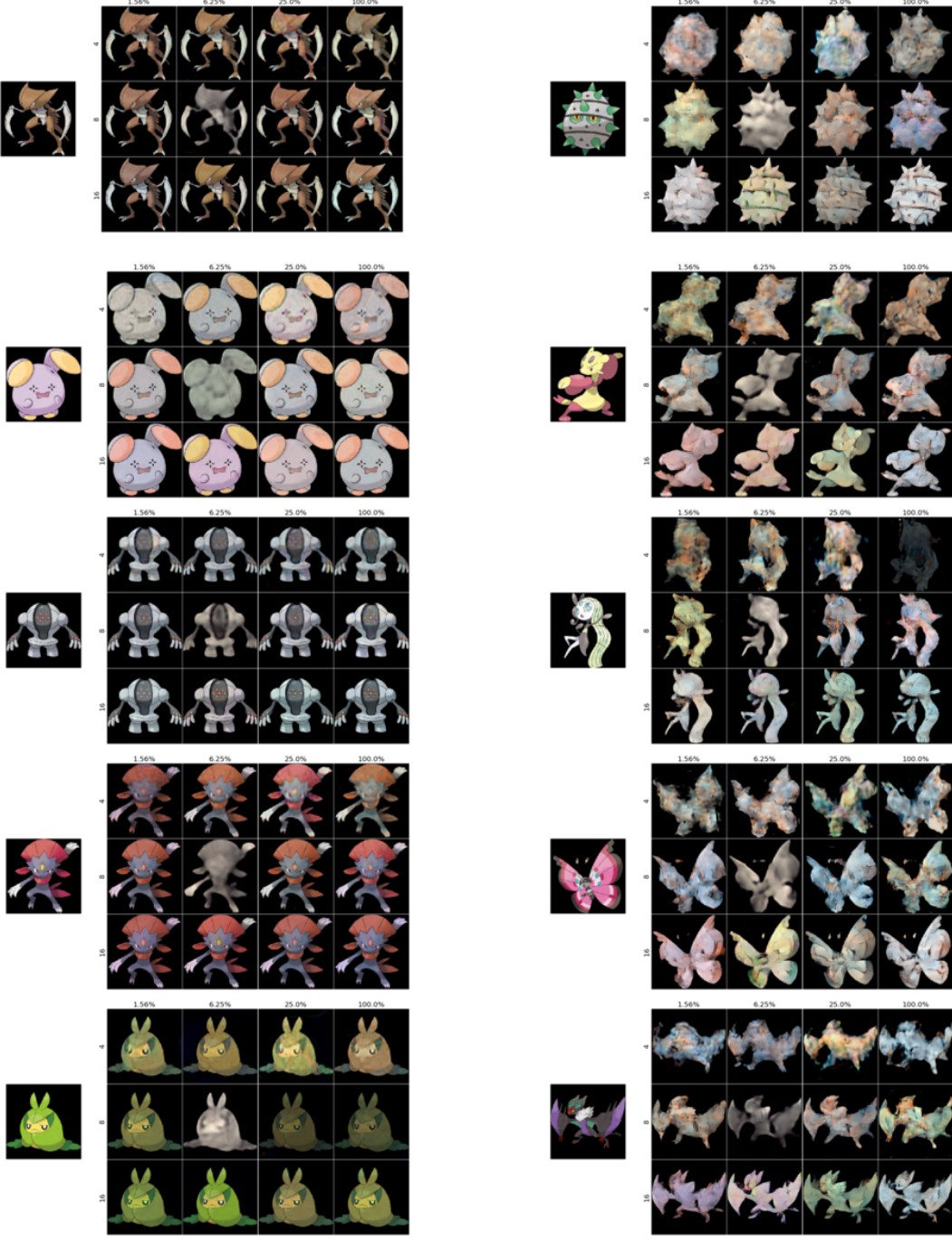

Figure A.1: Reconstructions of randomly picked samples from the Pokemon dataset. The left column contains samples from the training data, while on the right, we show samples from the test data. In each subfigure, the rows correspond to CAEs with the same bottleneck size (height, width), increasing from top to bottom. The columns group CAEs by the number of channels in the bottleneck, expressed as percentage relative to input given bottleneck size. The image to the left of each grid is the input image.

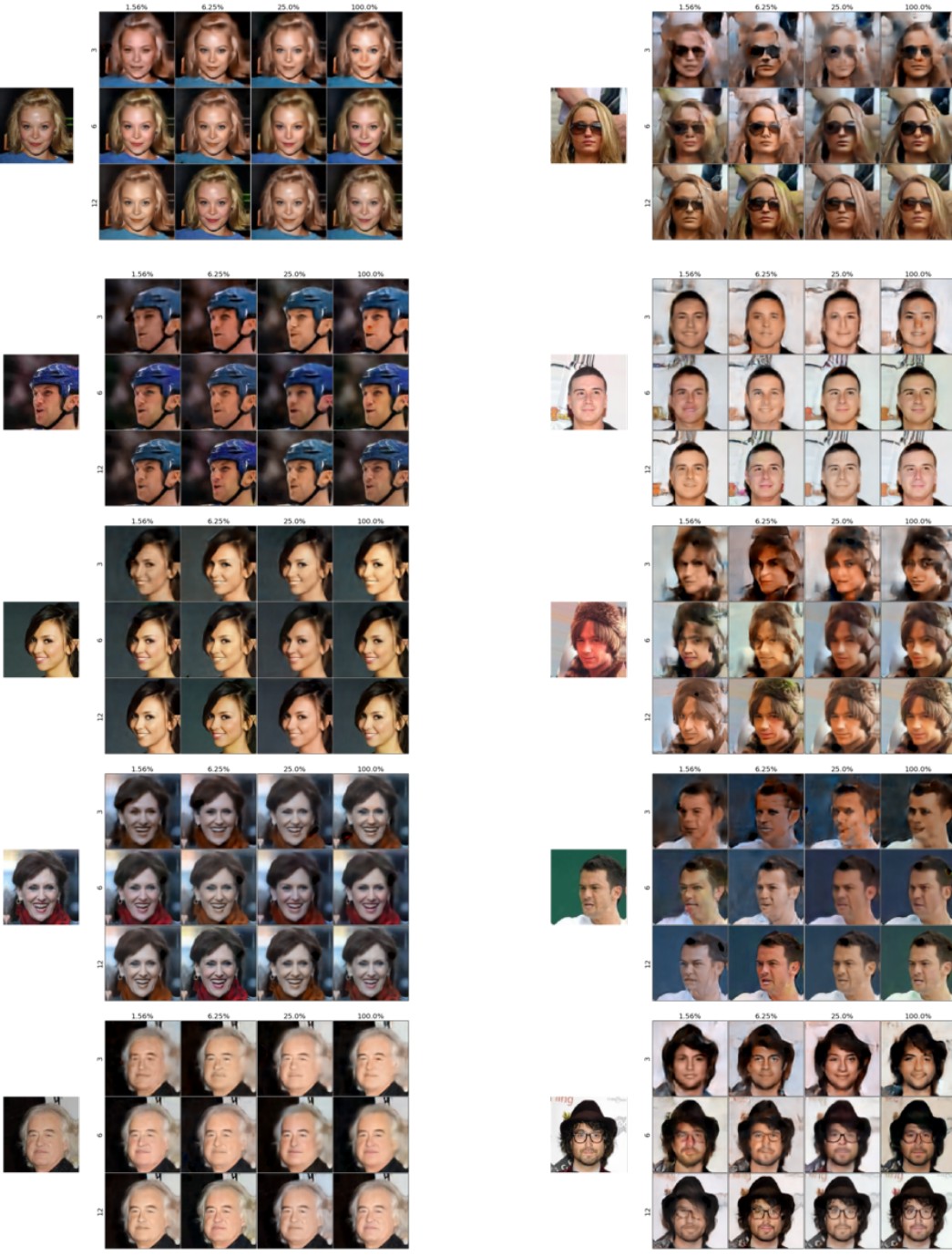

Figure A.2: Reconstructions of randomly picked samples from the CelebA dataset. The left column contains samples from the training data, while on the right, we show samples from the test data. In each subfigure, the rows correspond to CAEs with the same bottleneck size (height, width), increasing from top to bottom. The columns group CAEs by the number of channels in the bottleneck, expressed as percentage relative to input given bottleneck size. The image to the left of each grid is the input image.

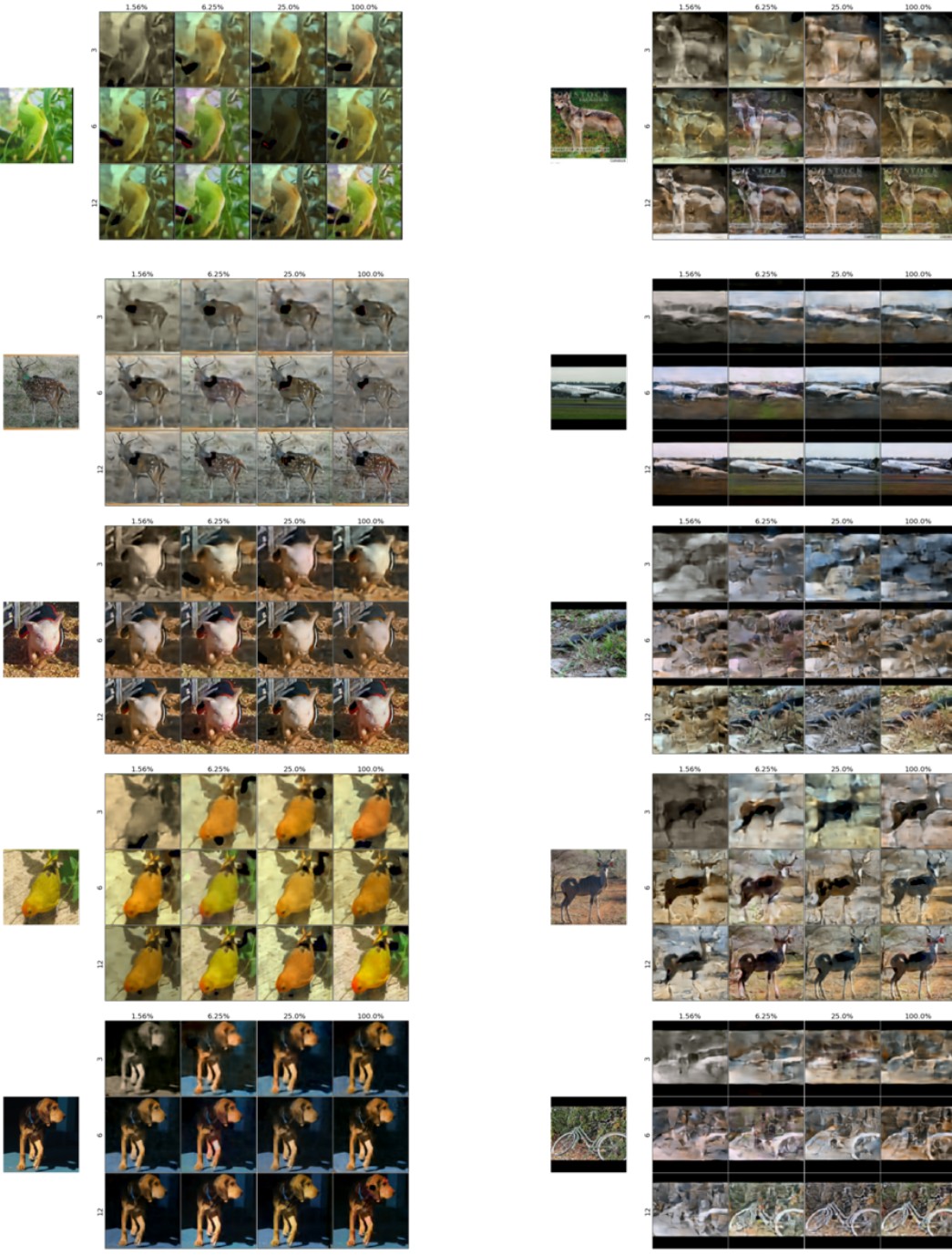

Figure A.3: Reconstructions of randomly picked samples from the STL-10 dataset. The left column contains samples from the training data, while on the right, we show samples from the test data. In each subfigure, the rows correspond to CAEs with the same bottleneck size (height, width), increasing from top to bottom. The columns group CAEs by the number of channels in the bottleneck, expressed as percentage relative to input given bottleneck size. The image to the left of each grid is the input image.

