# OpenReview forum: "Walking the Tightrope: An Investigation of the Convolutional Autoencoder Bottleneck"
_ICLR.cc/2020/Conference — Reject_

### Official Review · AnonReviewer3 · 2019-10-23
**Official Blind Review #3**

**Rating:** 3

**Review:**

Summary:
This paper studies some of the properties of fully convolutional autoencoders (CAE) as a function of the shape and total size of the bottleneck. They train and test CAEs with bottlenecks consisting of different ratios of spatial resolution versus number of channels, as well as different total number of neurons. The authors investigate which type of change in the bottleneck is most influential on training behavior, generalization to test set, and linear separability for classification/regression. Their first main finding is that the spatial resolution of the bottleneck is a stronger influencer of generalization to the test set than the number of channels and the total number of neurons in the bottleneck. The second main finding is that even when the total number of neurons in the bottleneck is equal to the data input size, the neural network does not appear to simply learn to copy the input image into the bottleneck.


Decision:
Weak reject: It is always refreshing to see papers that address/challenge/investigate common assumptions in deep learning. However, I find the experimental findings and discussion of borderline quality for a full conference paper. It might be more suitable as a good workshop paper.

Supporting arguments for decision:
It is unclear to me why the authors have chosen to only take a subset of the CelebA and STL-10 datasets for training and testing. It seems like dataset size is also an important factor that increases the complexity for training a model, and it certainly affects how a model can generalize. When auto-encoders are studied in other literature it is uncommon practice to restrict the dataset sizes like this, so this makes me question the applicability of this paper’s results to the literature.

It seems that the experimental validation is based on one run per CAE model with one single seed. This is on the low side of things, especially when quite extensive claims are made. An example of such a claim is on page 6 when discussing a sudden jump in training and test scores for 8x8x48 model trained on the Pokemon dataset. Because the same behavior appeared when the authors repeated the experiment with the same seed, the authors conclude “This outlier suggests, that the loss landscape might not always be as smooth towards the end of training, as some publications (Goodfellow et al., 2014) claim and that ‘cliffs” (i. e., sudden changes in loss) can occur even late in training.” Making this claim based on something that occurs with one single model for a single seed is not convincing and overstating this finding.
Another example is on page 7 under bullet point 4, where the authors discuss the obtained evidence against copying behaviour when the bottleneck is of the same size as the input. The authors state “We believe this finding to have far-reaching consequences as it directly contradicts the popular hypothesis about copying CAEs.” The paper definitely shows some empirical evidence that supports the claim that copying does not occur, but these findings are all done with a single seed and by considering small subsets of datasets (celebA and stl-10). In my opinion, it is therefore too much to state that the current findings have far reaching consequences. It has potential, but I wouldn’t go much further than that.

On page 7 in the second to last paragraph the influence of dataset complexity is discussed. The authors state “the loss curves and reconstruction samples do not appear to reflect the notion of dataset difficulty we defined in Section 2.3” and “This lack of correspondence implies that the intuitive and neural network definitions of difficulty do not align. Nevertheless, a more detailed study is required to answer this question definitively as curriculum learning research that suggests the opposite (Bengio et al., 2009) also exists.” It is unclear to me what the authors expected to find here. Moreover, the absence of major differences across the chosen datasets does not immediately make me doubt or question results from curriculum learning. My skepticism is again enhanced by the fact that the authors have taken a subset of the data for the more complex celebA and STL-10 datasets. Dataset size seems like a crucial part of dataset complexity.

An interesting smaller finding is that linear separability of the latent codes for classification is better on the test set for the pokemon dataset, even though the training and test reconstruction losses showed signs of overfitting. The authors hypothesize that overfitting might occur more in the decoder than in the encoder.

Additional feedback to improve the paper (not part of decision assessment)
- Section 2.3.1: what was the original resolution of the pokemon dataset?
- Section 3.1, c_j is used as the compression level, but in eq 3 and 4 c^i is also used to indicate the number of channels. - For clarity please use n_ci in eq 2 and 3 or choose a different all together for the compression level.
- Please increase the font size of the plots in figures 1, 3 and 4.


**Experience Assessment:**

I have published one or two papers in this area.

**Review Assessment: Checking Correctness Of Derivations And Theory:**

N/A

**Review Assessment: Checking Correctness Of Experiments:**

I carefully checked the experiments.

**Review Assessment: Thoroughness In Paper Reading:**

I read the paper thoroughly.

---

> ### Author Response · Authors · 2019-11-12
> **Response to Reviewer #3**
>
> Dear Reviewer #3,
> Thank you very much for the in-depth review. We are aware that you must have invested a lot of time. Thank you also for the many suggestions for improving our paper. We have re-written portions of our paper to address or clarify the cases of overstatements, which you pointed out. We will revisit those later in our response. However, we wish to start with the two most prominent points of criticism we received from you.
>
>
> "It is unclear to me why the authors have chosen to only take a subset of the CelebA and STL-10 datasets for training and testing."
>
> An important question we investigated was how and when CAEs overfit and what role the bottleneck shape plays in this process. From our experience prior to conducting this investigation, we knew that CAEs do not seem to overfit in typical training scenarios. Therefore, to be able to observe overfitting, we deliberately limited the dataset size and increased the number of epochs to 1000.
> Training on more data certainly affects the generalization of the model. However, we argue that the observed pattern remains unchanged, which is what we observe when going from the small Pokemon dataset to CelebA and STL-10: the generalization gap narrows but the overall trends remain the same
> We are not sure which other literature you are referring to when stating, "When auto-encoders are studied in other literature, it is uncommon practice to restrict the dataset sizes like this." If it serves the research question, we believe that limiting the dataset size is a viable, investigative tool, e.g., as used in [1]. We agree, however, that we were not clear about our reasoning in the paper, and thank you for pointing it out. Thus, we have slightly modified the dataset section in the revision to correct this.
> We agree with your statement that "Dataset size seems like a crucial part of dataset complexity." What we wanted to investigate, however, is how the complexity of the data samples influences the CAE training. We also did not always clearly communicate this and have tried to fix this in the revision.
>
>
> "It seems that the experimental validation is based on one run per CAE model with one single seed."
>
> We understand your concern about the statistical significance of the results. However, we set up our experiments in a way that minimizes the number of moving parts and, therefore, the risk of confounding. By fixing the seed, we perform an intervention experiment to find the influence of the bottleneck shape on the CAE. Furthermore, we experimented on three different datasets and observed the same outcome every time, which, we believe, gives credibility to our results. Of course, we would have liked to repeat the experiments for other seeds but, unfortunately, our resources are limited. Because we wanted the bottleneck to be the only moving part, we had to preserve the total number of pixels in the input until the bottleneck layer (as explained in the methods section). This constraint resulted in massive models, the smallest of which had ~5x10⁷ parameters (the largest had ~9x10⁷). These models are on the order of ResNet152, which has ~6x10⁷ parameters. The training duration for a single model is on the order of days, and we trained 36 models in total, excluding the classifiers for the knowledge transfer experiments. Since the submission deadline, we have been training more models with a different seed, but so far, only 30 out of the 36 models could be trained. So far, we observe exactly the same pattern. Unfortunately, training will not finish before the revision deadline. Should our paper be accepted, we will, of course, include those results in the camera-ready version. However, we argue that our results show a level of significance and novelty that warrants publication, to inspire other researchers to replicate and expand upon them and hence deepen our understanding of CAEs and perhaps CNNs in general. To this end, we have published all of our code, models, and data, as mentioned in the paper.
> We understand that perhaps other authors share our struggle for resources, as many outstanding autoencoder-related papers from last year's ICLR also presented an analysis based on a single initialization. See [2], [3], [4] and [5] for examples.
>
>
> Lastly, we wish to thank you for highlighting the points where you believe we overstated our results. Our intention was not to question the works related to the loss landscape in DNNs and curriculum learning. We may have phrased these particular passages poorly, which is why we have changed them in the revised version. Regarding the 8x8x48 model, we wanted to express our surprise, that the model parameters underwent such a drastic change late in training and also hint at the possibility that there is more to discover here. The same goes for the part about curriculum learning.
>
> Overall, we thank you again for your consideration and hope we have addressed your concerns. If so, we would be happy if you re-evaluated our paper.

---

> > ### Comment · AnonReviewer3 · 2019-11-14
> > **Response to rebuttal**
> >
> > Thank you for the extensive rebuttal.
> >
> > First, I appreciate that the authors have toned down their claims a little, and I of course understand that resources can be scarce. Please mention the model sizes in the revised vision of your paper.
> >
> > I am still concerned about the lack of results on the influence of dataset size, specifically when looking at this separately from the dataset image complexity.
> > I understand that the question “how does the bottleneck size and shape influence overfitting” is what you want to investigate. My concerns/questions remain two-fold: i) Is it enough to only focus on image complexity when studying overfitting and the influence of the bottleneck? As overfitting is strongly influenced by dataset size, this seems like something that can, and in my opinion should, be studied more here. ii) As you state “From our experience prior to conducting this investigation, we knew that CAEs do not seem to overfit in typical training scenarios.” This then raises the question whether focusing on a scenario of overfitting, which doesn’t happen so often, is actually impactful enough for a full conference paper as it is unclear how the findings translate to “typical training scenarios”.
> >
> > Experiments where the subset size of the dataset is changed would give more insight into how the findings in this paper can be extended to “typical training scenarios”. I agree that when looking at overfitting, the complexity of the dataset in terms of image complexity is interesting.
> > But when overfitting is studied, dataset size is also a critical component. As you say, confounding factors should be avoided. Stating that the trends remain the same when switching from the smaller pokemon dataset to a subset of CelebA/STL-10 (which differ both in image complexity as well as the dataset size) does not provide sufficient evidence (in my opinion) that the findings in the paper will transfer to scenarios where for instance the entire celeba/stl-10 datasets are considered.
> >
> > With my statement "When auto-encoders are studied in other literature, it is uncommon practice to restrict the dataset sizes like this." I am referring to other autoencoder papers (such as [1, 2, 3]) that also study celeba/stl-10, but don't restrict the size like you do. If you would show results where the size restriction is varied (including the full dataset), it would make the paper more convincing.
> >
> > I want to emphasise that I don't want to discourage the authors from attempting to publish this. I simply think that in its current form it's not impactful or complete enough to be a full conference paper.
> >
> > [1]Wasserstein Auto-Encoders. Tolstikhin et al.  https://arxiv.org/abs/1711.01558
> > [2] beta-VAE: Learning Basic Visual Concepts with a Constrained Variational Framework . Higgins et al. https://openreview.net/forum?id=Sy2fzU9gl
> > [3]Autoencoding beyond pixels using a learned similarity metric. Larsen et al. https://arxiv.org/abs/1512.09300

---

> > > ### Author Response · Authors · 2019-11-15
> > > **Response to response to rebuttal :)**
> > >
> > > Thank you for replying to our rebuttal and continuing to participate in the discussion. We much appreciate the feedback we get from you!
> > >
> > > Varying the dataset size is a great idea, and we will definitely investigate it in future work. However, we want to make a few clarifications regarding this submission:
> > >
> > > How and when CAEs overfit is one of the questions we try to address in our paper. However, it is only part of our investigation. In this submission, we offer a broad investigation of CAE characteristics: Effect of bottleneck shape on training dynamics, knowledge transfer and representation structure, testing of the copying CAE hypothesis, and overfitting behavior. We find all of these aspects to be underreported in literature. The sources you provide are an excellent example of this. They mostly provide qualitative results, focused on generative elements of the model, while completely lacking a description of the aspects we investigated in our submission. Only the WAE paper mentions that the model was trained for 55 epochs in the appendix, while the others do not disclose information about the training of the model. We want to emphasize that we do not criticize the quality of these papers but rather comment on the current focus of AE literature in general.
> > >
> > > We understand you are concerned about the results not replicating when the models are trained on the full dataset. However, we do not believe this outcome to be likely. Our reasoning for this is two-fold:
> > > i) Studies, such as  Zhu et al. Do We Need More Training Data? (https://arxiv.org/pdf/1503.01508.pdf) have shown that adding more data quickly results in diminishing returns. Therefore, we do not expect generalization to increase to a point where all models achieve equal performance on the test data, even when we increase the number of samples.
> > > ii) The pattern we are reporting, regarding the effect of bottleneck shape, manifests itself early in training, both in training and test loss. At this stage, the model has clearly not captured all the variance present in the training data. We do not see how increasing the dataset size would drastically change this behavior.
> > > Additionally, many image datasets in other domains, such as medical imaging, do not have 100k+ samples, although they use CAEs frequently, especially for tasks like outlier detection. We, therefore, argue that limiting the dataset size to 10k samples is representative of what can we encounter in many domains.
> > >
> > > Lastly, when I stated that “From our experience prior to conducting this investigation, we knew that CAEs do not seem to overfit in typical training scenarios.” I should have been more careful with the wording. "Suspected" would have been more appropriate (I am definitely learning about the danger of overstatement from this submission). I was referring to experiments, which we performed on medical imaging data where the validation error did not increase even when trained for over 1000 epochs on datasets with 3-6k samples (full public datasets) (here is a link to the figure https://ibb.co/FbbbLr4). This observation motivated the question as stated in this submission: "How and when do CAEs overfit?".  To the best of our knowledge, the literature does not discuss this aspect at all. Therefore, we argue that our observations of the overfitting behavior (or apparent lack thereof for larger feature maps in the bottleneck) already provide novel insights into the workings of CAEs.

---

> > > > ### Comment · AnonReviewer3 · 2019-11-15
> > > > **Another response**
> > > >
> > > > Thank you for your response.
> > > >
> > > > I understand that you think your findings will likely transfer to full dataset training. I just want to see more concrete evidence.
> > > >
> > > > If you think studying small datasets is more relevant to other domains such as medical imaging, then please show results on medical imaging datasets. It is difficult to imagine how your findings translate to such a setting, as medical images have very different image characteristics as compared to CelebA or STl-10, and often have a fairly high resolution.

---

> > > > > ### Author Response · Authors · 2019-11-15
> > > > > **Results from different seed**
> > > > >
> > > > > Unfortunately, we will not be able to address your criticism of the limited dataset set size in this submission. We thank you for the valuable discussion and will incorporate your feedback into our future work.
> > > > >
> > > > > Regarding your other main point of criticism: As we stated earlier, we have been training more models since the submission deadline. We have managed to temporarily acquire additional hardware resources to speed up training of the remaining models on a different seed. We have included the results in the most recent revision of the paper. The plots for models 12x12x48 and 12x12x192 for STL-10 stop at about 850 epochs, because they are still finishing their training. Nevertheless, you will find that the pattern for this seed remains the same as for the first. We hope that this increases your confidence in our results.
> > > > >
> > > > > We thank you again for taking so much time to review our paper and discussing it with us.

---

> ### Author Response · Authors · 2019-11-12
> **Response to Reviewer #3 (Sources)**
>
>
> [1] Radhakrishnan, A., Yang, K., Belkin, M., & Uhler, C. (2019). Memorization in Overparameterized Autoencoders. ICML 2019 Deep Phenomena.
> [2] Berthelot, D., Raffel, C., Roy, A., & Goodfellow, I. (2018). Understanding and Improving Interpolation in Autoencoders via an Adversarial Regularizer.
> [3] Gregor, K., Papamakarios, G., Besse, F., Buesing, L., & Weber, T. (2018). Temporal Difference Variational Auto-Encoder.
> [4] Kolouri, S., Pope, P. E., Martin, C. E., & Rohde, G. K. (2018). Sliced Wasserstein auto-encoders.
> [5] Li, X., Chen, Z., Poon, L. K., & Zhang, N. L. (2018). Learning Latent Superstructures in Variational Autoencoders for Deep Multidimensional Clustering.

---

### Official Review · AnonReviewer1 · 2019-10-24
**Official Blind Review #1**

**Rating:** 6

**Review:**

The authors evaluate convolutional autoencoders (CAE) by varying the size (width & height) and depth of the bottleneck layer on three datasets and compare test and training performance. They furthermore evaluate the quality of the bottleneck activations for linear classification. The authors also investigate the belief that a bottleneck layer of size equal to the input image will copy the image.

I am not an expert in the field of (C)AEs. As such, I cannot approriately judge the relevance of the questions which are answered here. In the following, I will therefore make the assumption that those questions are relevant.
If so, I (weakly) recommend accepting the paper.

While it does not propose any novel algorithms, it does ask a clear question and provides a compelling experimental answer, which (assuming the question is relevant), should be interesting for the community.
On the other hand, further experiments to provide initial insights into the further questions raised by the authors would improve the 'novelty' aspect of the paper.

Minor comment:
I believe the paper "Reconciling modern machine learning practice and the bias-variance trade-off" by Belkin et. al is relevant.

Edit:
Thank you for your response.

I wasn't referring to any specific experiments, but the questions you are raising in the paper, for example in the conclusion section.

Regarding Belkin et al: Sorry for the misunderstanding, I didn't mean to imply that it's highly relevant, but I thought it might be a good addition to the related work section as it also relates to the generalization of neural networks.


**Experience Assessment:**

I do not know much about this area.

**Review Assessment: Checking Correctness Of Derivations And Theory:**

N/A

**Review Assessment: Checking Correctness Of Experiments:**

I assessed the sensibility of the experiments.

**Review Assessment: Thoroughness In Paper Reading:**

I read the paper at least twice and used my best judgement in assessing the paper.

---

> ### Author Response · Authors · 2019-11-12
> **Response to Reviewer #1**
>
>
> Dear Reviewer #1,
>  thank you very much for appreciating our work.
>
>  The novelty in our work lies in the systematic investigation of commonly held beliefs about CAEs. As far as we know (please correct us if we are wrong), these beliefs have never been thoroughly tested, despite their wide-spread adoption. As you have observed, we tried to formulate these beliefs as testable hypotheses and verify them experimentally. In the review, you state that "further experiments to provide initial insights into the further questions raised by the authors would improve the 'novelty' aspect of the paper." Could you please provide specific experiments that you would like to see added to the paper? We will try our best to include them, if possible, before the revision deadline.
>
>  The paper you mention in the review is fascinating, although probably not closely related to our work. Belkin et al. investigate how increasing the number of model parameters past the number of training samples increases generalization. In our case, the smallest model that we trained already had ~5x10⁷ parameters due to the experiment setup, which is far beyond the "interpolation point" mentioned in Belkin et al. Additionally, we demonstrate an increase in generalization while keeping the number of parameters fixed. For example, the models 3x3x48, 6x6x48, and 12x12x48 have the same number of parameters, yet their generalization is significantly different. Please let us know if we missed something.

---

### Official Review · AnonReviewer2 · 2019-10-25
**Official Blind Review #2**

**Rating:** 3

**Review:**

This paper investigates convolutional autoencoder (CAE) bottleneck. The research problem is important, as CAE is widespread adopted. It cam be interesting to shed insight into how the bottleneck works.

In particular, two observations are made: (1) By measuring the performance of the latent codes in downstream transfer learning tasks, the authors show that  increased height/width of the bottleneck drastically improves generalization; The number of channels in the bottleneck is secondary in importance.  (2) CAEs do not learn to copy their input, even when the bottleneck has the same number of neurons as there are pixels in the input.

It would make this submission more convincing if the authors show the some important applications of their findings. For example,

A. How does (1) above help architecture search when designing CAEs in a new application?
B. How does (2) above help style transfer if ``"CAEs do not learn to copy their input"

Concerns: If (2) is true, Could the authors explain that almost perfect reconstruction results are shown in [*] ?

[*] Generating Diverse High-Fidelity Images with VQ-VAE-2

**Experience Assessment:**

I have read many papers in this area.

**Review Assessment: Checking Correctness Of Derivations And Theory:**

N/A

**Review Assessment: Checking Correctness Of Experiments:**

I assessed the sensibility of the experiments.

**Review Assessment: Thoroughness In Paper Reading:**

I read the paper at least twice and used my best judgement in assessing the paper.

---

> ### Author Response · Authors · 2019-11-12
> **Response to Reviewer #2**
>
> Dear Reviewer #2,
> thank you very much for investing the time to review our paper and providing suggestions for improvement, which we tried to incorporate into our work.
>
>
> "A. How does (1) above help architecture search when designing CAEs in a new application? ":
>
>  The most considerable areas of application for autoencoders, outside of representation learning in academia, are outlier detection, and image compression and denoising. Our results are readily applicable to both of these domains: In the case of outlier detection, the model should yield a high reconstruction error on out-of-distribution samples. Using smaller bottleneck sizes to limit generalization could prove useful in this scenario. Compression and denoising tasks, on the other hand, seek to preserve image details while reducing file size and discarding unnecessary information, respectively. In this case, a bigger bottleneck size is preferable, as it increases reconstruction quality at the same level of compression.
> Additionally, our paper demonstrates that CAE without a bottleneck does not learn simply to copy the input. Although further investigation is needed to determine what is preventing the CAEs from doing so, we believe that identifying the source of this phenomenon could lead to new architectures and training criteria that improve the abstraction capabilities of the CAE. We argue that the work we presented here is a starting point for such an investigation. We recognize that we did not communicate these implications clearly, and thank you for bringing them to our attention. We have re-written a portion of the discussion and conclusion in our paper to incorporate this aspect.
>
>
> "B. How does (2) above help style transfer if ``"CAEs do not learn to copy their input." "
>
> Currently, we are not aware of CAEs being used in the field of style transfer. To the extent of our knowledge, style transfer mostly relies on a pre-trained VGG, but we have to admit, we are no experts in this field. Could you perhaps provide a specific example for the use of CAEs in style transfer? This way, we could potentially offer insight into how our work is related.
>
>
> "Concerns: If (2) is true, could the authors explain that almost perfect reconstruction results are shown in [*] ?"
>
> Thank you for pointing us to this paper. It was an interesting read. In the article, the authors use a combination of two representations, one is 64x64, and the other is 32x32, whereas the original input size is 256x256. First of all, copying is, by definition, not possible in this scenario, as the number of neurons in the bottleneck is smaller than the number of pixels in the input. The high-fidelity reconstruction presented in this paper nicely aligns itself with our findings. As we demonstrate in our paper, the reconstruction improves with increased bottleneck size (as in size of the feature maps). In our experiments, we only tested up to a bottleneck size of 16x16 with an original image size of 128x128. Extrapolating our findings, we would expect the reconstruction quality to improve further for bigger bottleneck sizes, which can be seen in Figure 3 of [*] when they include the larger feature maps.
> Furthermore, they include a sophisticated system for incorporating hierarchical features into the architecture, which presumably also benefits abstraction and in turn, better reconstruction. Additionally, one can formulate the hypothesis, that the bottleneck size in [*] is so big that the task of the decoder is potentially more akin to 4x super-resolution than to decoding, and many papers have demonstrated the power of CNNs in this setting (e.g., Wang 2018). However, this is not investigated in the article, so a separate experiment is necessary to verify this.
>
> Wang, Xintao, et al. "Esrgan: Enhanced super-resolution generative adversarial networks." Proceedings of the European Conference on Computer Vision (ECCV). 2018

---

### Decision · Program_Chairs · 2019-12-19

**Decision:**

Reject

**Comment:**

The paper investigates the effect of convolutional information bottlenecks to generalization. The paper concludes that the width and height of the bottleneck can greatly influence generalization, whereas the number of channels has smaller effect. The paper also shows evidence against a common belief that CAEs with sufficiently large bottleneck will learn an identity map.

During the rebuttal period, there was a long discussion mainly about the sufficiency of the experimental setup and the trustworthiness of the claims made in the paper. A paper that empirically investigates an exiting method or belief should include extensive experiments of high quality in to enable general conclusions. I’m thus recommending rejection, but encourage the authors to improve the experiments and resubmitting.